# Variability of functional and biodiversity responses to perturbations is predictable and informative

James A. Orr ®[1,2] ✉, Jeremy J. Piggott ®[3], Andrew L. Jackson ®[3], Michelle C. Jackson ®[1] & Jean-François Arnoldi ®[4]

Perturbations such as climate change, invasive species and pollution, impact the functioning and diversity of ecosystems. However diversity has many meanings, and ecosystems provide a plethora of functions. Thus, on top of the various perturbations that global change represents, there are also many ways to measure a perturbation's ecological impact. This leads to an overwhelming response variability, which undermines hopes of prediction. Here, we show that this variability can instead provide insights into hidden features of functions and of species responses to perturbations. By analysing a dataset of global change experiments in microbial soil systems we first show that the variability of functional and diversity responses to perturbations is not random; functions that are mechanistically similar tend to respond coherently. Furthermore, diversity metrics and broad functions (e.g. total biomass) systematically respond in opposite ways. We then formalise these observations to demonstrate, using geometrical arguments, simulations, and a theory-driven analysis of the empirical data, that the response variability of ecosystems is not only predictable, but can also be used to access useful information about species contributions to functions and population-level responses to perturbations. Our research offers a powerful framework for understanding the complexity of ecological responses to global change.

Describing aggregate properties of ecosystems and predicting their behaviour in the face of perturbations is a major goal of contemporary ecology. If consistent patterns emerged when considering aggregate-level responses, ecologists could aim for data-based predictions and provide clear, practical recommendations[1,2]. However, there are many relevant aggregate properties to consider, from diversity metrics to ecosystem functions, that may all respond in different ways to perturbations[3,4]. As there is no obvious way to organize this variability, the hopes for general predictions of community-level responses to perturbations can seem slim.

The importance and origin of species diversity was a central theme of late 20th century ecology[5–8], which led to a proliferation of metrics to define and measure diversity based on the richness, evenness and rarity of species[9–13]. Since then, understanding how species collectively perform a function has become a prominent area of research[14–17], with clear implications for our understanding of concrete issues regarding productivity, carbon sequestration, pollination, or nutrient cycling of natural or engineered ecosystems. In light of rapid anthropogenic global change, there is currently increased focus on understanding how aggregate ecological properties will respond to

[1]Department of Biology, University of Oxford, Oxford, UK. [2]School of the Environment, University of Queensland, Brisbane, QLD, Australia. [3]Zoology, School of Natural Sciences, Trinity College Dublin, Dublin, Ireland. [4]Centre National de la Recherche Scientifique, Experimental and Theoretical Ecology Station, Moulis, France. ✉e-mail: james.orr@uq.edu.au

perturbations such as land-use change, invasive species, climate change and pollution[4,18–20].

Ecologists are very aware that different aggregate properties, such as diversity metrics or ecosystem functions, describe very different aspects of communities and may thus respond in completely different ways to a given environmental perturbation[3,4,21]. For instance, the many different diversity metrics employed by ecologists describe different facets of community structure[22,23]. If a perturbation caused the extinction of rare species while making the overall abundance distribution of the community more even, species richness would decrease, but a measure of evenness (e.g. Simpson's index) would increase.

Similarly, ecosystem functioning takes many forms, and can be measured in a myriad of ways. Some functions, such as biomass production or respiration, are *broad* functions: they are performed by most or all species in a community. Other functions, such as the breakdown of specific chemicals or the production of specific enzymes, are *narrow* in the sense that they require the presence of particular species, or combinations of species, to be performed[24,25]. The great variety of ecosystem functions—in what they do, how broad or narrow they are, how species contribute to them, and how they respond to perturbations—has motivated the rapid development of multifunctional ecology where multiple functions are considered at once to more accurately characterize the state of an ecosystem[26–28].

In the face of this inherent ecological complexity, what can be learned from the variability of functional and biodiversity responses to perturbations? Here we claim that this variability can be used to explore hidden features of ecosystems and of perturbations.

To make this point we analyse data from global change experiments conducted in microbial soil systems (Box 1 and Fig. 1). Focusing on three diversity metrics, two broad ecosystem functions, and eight narrow ecosystem functions, we explore patterns of mismatches between functional and diversity responses to global change factors (such as pollution, environmental events or land-use change, all seen here as perturbations). Concretely, we look at the proportion of cases where one aggregate property responds negatively to a perturbation while the other responds positively to it. As expected, we find a great degree of variability in responses to perturbations. This variability, however, is not random, but instead shows a recognizable degree of structure. Aggregate properties that are thought to describe ecosystems in similar ways (e.g. production of beta-xylosidase and production of cellobiohydrolase, enzymes that contribute to carbon cycling) have a lower proportion of mismatches than would be expected by chance (modules of blue squares, Fig. 1A). On the other hand, diversity metrics and ecosystem functions tend to systematically differ in how they respond to perturbations (dominance of red squares between diversity and

## BOX 1
# Initial analysis of empirical data

To quantify the variability of functional and biodiversity responses to perturbations we analysed a dataset of global change experiments conducted in microbial soil systems[20]. This dataset contained 1235 perturbations from 341 publications. Perturbations included warming, elevated carbon dioxide levels, altered precipitation, nutrient enrichment, land-use change, or combinations of these factors. The effect of each perturbation in a given experiment was quantified using the natural logarithm-transformed response ratio:

$$RR = \ln\left(\frac{X_t}{X_c}\right) \qquad \text{(Box 1 Eq 1)}$$

where $X_t$ and $X_c$ are the means of the treatment and control groups for a given aggregate property. The variances of these effect sizes are also available in the dataset, but we do not require them for this initial analysis as we do not exclude points based on some statistical cutoff. Indeed, following our geometric approach, there is no reason to expect that the proportion of mismatches between two aggregate properties would be different for data points with or without statistically significant results.

Each individual perturbation was quantified using multiple aggregate properties covering a wide range of ecosystem functions and measures of diversity. We focused on aggregate properties where all pairs had at least ten observations in the dataset so that the proportion of mismatches between them could be estimated with some robustness. This arbitrary number of observations was chosen to strike a balance between having enough observations to estimate proportions of mismatches reliably and having enough pairs of aggregate properties to see general patterns across broad functions, narrow functions, and diversity metrics. Choosing other cut-offs does not qualitatively change the results (demonstrated in the R markdown at https://doi.org/10.5281/zenodo.13985015). This filtering of the data returned 1015 perturbations that were measured with at least two of thirteen aggregate properties including three measures of diversity (richness, Shannon index, and Chao index), two broad ecosystem functions (biomass and respiration), and eight narrow ecosystem functions subdivided into P-cycling enzymes (phosphatase), N-cycling enzymes (*N*-acetyl-beta-glucosaminidase), hydrolytic C-cycling enzymes (beta-xylosidase, cellobiodydrolase, beta-glucosidase, and alpha-glucosidase), and oxidative C-cycling enzymes (peroxidase, phenol oxidase). Details of how the functions were measured (e.g. whether respiration was calculated in the laboratory or the field, or whether enzymes were measured using colourimetric or microplate assays) can be found in Zhou et al. (2020).

This list of aggregate properties was sorted a priori based on intuitions about their underlying mechanisms (grouped by diversity metrics, broad functions and narrow functions based on Zhou et al. (2020)) and a heatmap was made to visualize the proportion of mismatches between each pair (Fig. 1A). If the variability between aggregate properties was just random (i.e. if the heatmap was all white or just showed random distributions of red and blue) there might not be much more to say, but if the heatmap showed some structure there could be useful information to gain from the variability. Indeed, the modularity of the heatmap shows that aggregate properties that are thought to be similar tend to respond to perturbations similarly (e.g. relatively low proportion of mismatches—ranging from 0.16 to 0.28—between measures of diversity). Conversely, groups of aggregate properties that describe different aspects of a community can systematically differ in their responses to perturbations (e.g. an abundance of red between diversity metrics and ecosystem functions, with the proportion of mismatches going as high as 0.73).

We will return to these empirical results after we have outlined our geometrical approach for quantifying the notion of similarity between aggregate properties. In fact, we can use our framework to reinterpret these empirical data to gain useful insights into how the perturbations in these experiments impacted these communities and also into how the species in these communities contribute to the different ecosystem functions.

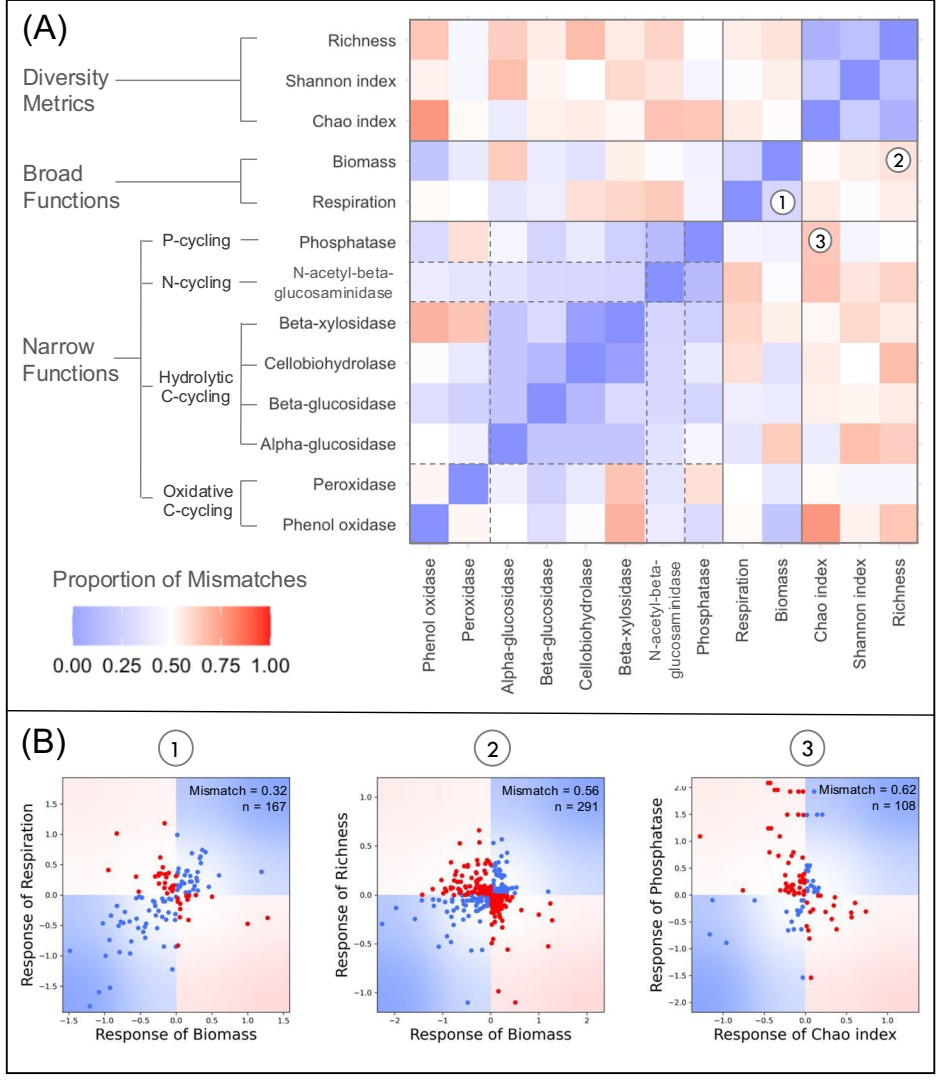

**Fig. 1 | Variability of functional and biodiversity responses to global change factors in experimental microbial soil systems. A** The proportion of qualitative mismatches (one responds positively and the other negatively) between thirteen aggregate properties including three measures of diversity (richness, Shannon index, and Chao index), two broad ecosystem functions (biomass and respiration), and eight narrow ecosystem functions subdivided into P-cycling enzymes (phosphatase), N-cycling enzymes (N-acetyl-beta-glucosaminidase), hydrolytic C-cycling enzymes (beta-xylosidase, cellobiodyhydrolase, beta-glucosidase, and alpha-glucosidase), and oxidative C-cycling enzymes (peroxidase and phenol oxidase). **B** Correlations between the responses of three specific pairs of aggregate properties: (1) biomass and respiration, (2) biomass and richness, and (3) Chao index and phosphatase. Points that fall in the blue areas of the plots were cases when the two metrics responded in the same way to a perturbation in a given experiment, while points that fall in the red areas were cases when there were qualitative mismatches between observations. The proportion of points that are red in these figures corresponds to the proportion of mismatches reported in (**A**).

ecosystem functions, Fig. 1A). Our intuitions about how mechanistically similar aggregate properties are (i.e. how we ordered the observations Fig. 1A) thus provide a useful starting point for understanding ecosystem's response variability, and we also find that there exist generic diversity-function response patterns.

Motivated by the findings of this empirical synthesis, we propose a framework that helps us glean useful, hidden information from the variability of functional and diversity responses to perturbations. To do so, we convert the ecological problem into a simpler geometrical one by representing perturbations as displacement vectors and community aggregate properties as directions in community state-space (the high-dimensional space whose axis reports the biomass of all constituent species). The central ingredient of our framework is a geometrical definition of collinearity between two aggregate properties which quantifies their similarity and predicts whether they will respond to a perturbation in the same way (Fig. 2). This prediction

assumes a high response diversity at the species level, and depends on how species' responses to perturbations scale with their biomass. Here, coarse-grained assumptions about population-level responses are used to better understand ecosystem functions. Conversely, we show that with some knowledge of the aggregate properties used to observe the ecological impacts of perturbations, the variability of these observations can be leveraged to gain information about species response diversity and how species' responses scale with their biomass. Armed with our geometrical framework we then reanalyse the empirical data from microbial soil systems to gain new insights on soil microbial ecosystem functions and how they are being impacted by anthropogenic global change. As well as proposing novel methods for validating and applying our framework to ecological data (outlined in an online tutorial at https://jamesaorr.github.io/community-properties-tutorial), we more broadly aim to inspire new approaches to studying complex ecological systems that embrace the variability of

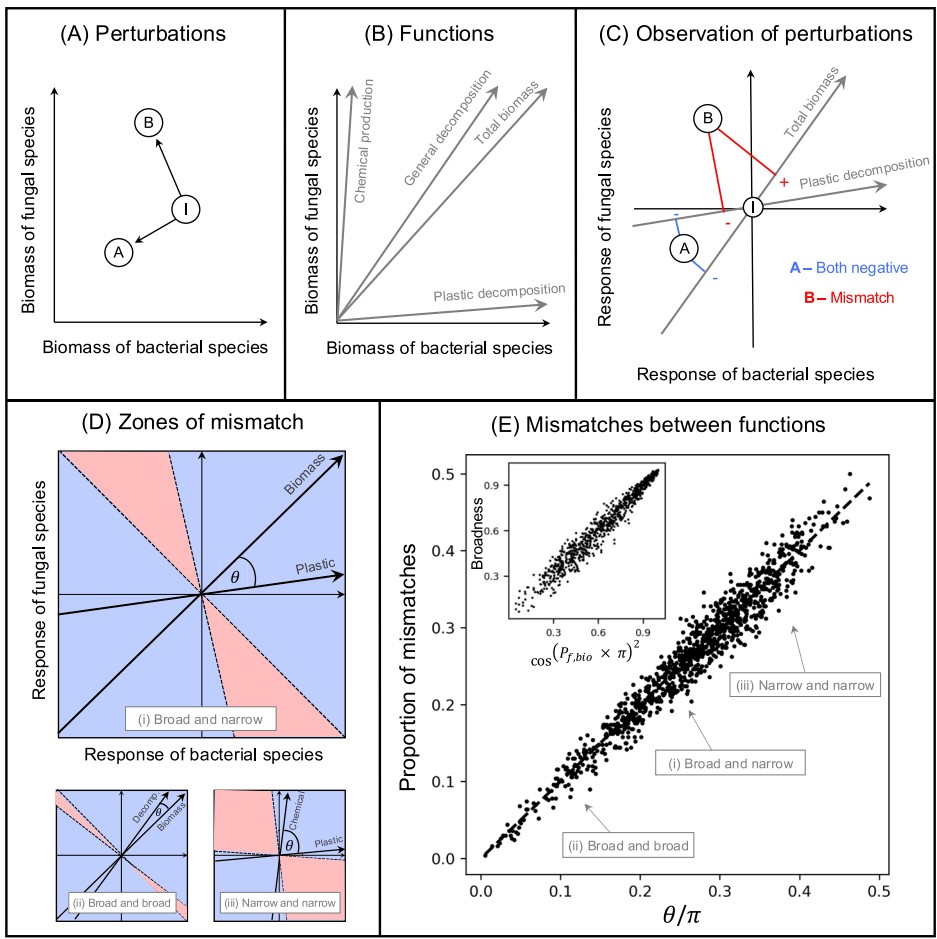

**Fig. 2 | Geometrical approach for relating the collinearity of functions to mismatches in their observations of perturbations. A** Perturbations can be viewed as displacement vectors in community state space. Here a hypothetical community of a bacterial species and a fungal species is independently impacted by two perturbations, represented by the black arrows from the initial state of the community (*I*) to the points *A* and *B*. **B** Measures of ecosystem function can be represented as positive directions in this state-space. **C** Perturbed states are plotted in a space where the initial state of the community is at the origin and each axis describes the response of each species to a perturbation. Here the displacement vectors associated with the independent perturbations *A* and *B* from (**A**) are projected onto the directions representing total biomass and plastic decomposition. For *A*, both functions observe negative responses. However, for *B* there is a mismatch in the observations of the functions: total biomass responds positively while plastic decomposition responds negatively. **D** For two functions, the zones of mismatches in their observations can be found by drawing lines perpendicular to the functions that go through the origin. Aggregate properties will observe different responses for perturbations that fall between these lines (i.e. in the red zones). The angle between the two functions determines the size of the zones of mismatches. The three pairs of functions include (i) a broad and a narrow function, (ii) two broad functions, and (iii) two narrow functions. **E** Over many in silico perturbation experiments, the proportion of mismatches between functions can be predicted by the angle between them in radians ($\theta$) divided by the number $\pi$. Two broad functions (ii) would have high collinearity and a low proportion of mismatches, one broad and one narrow function, (i) would have moderate collinearity and a moderate proportion of mismatches, while two narrow functions, and (iii) would have low collinearity and a high proportion of mismatches (unless they were performed by the same species). The inset shows that the proportion of mismatches between a function and total biomass is a very good predictor of the broadness of that function.

community-level responses to perturbations, using perturbations as probes to reveal hidden features of ecosystem dynamics and functioning.

## Results

Our geometrical arguments—outlined in the Methods and in Fig. 2—are well supported by our simulation results and can be used to refine the analysis of the empirical data. By simulating perturbation experiments on species-rich communities, we show how mismatches in the observations of two functions can be used to quantify the similarity of those functions and can be used to estimate a notion of response diversity (Fig. 3). We also found that mismatches in the observations of a function and a diversity metric can be used to quantify the scaling of perturbations by species biomass (Fig. 4). Returning to the empirical data, we applied a validation test (formally described in Supplementary Note 2) to show that the data, when grouped by biome, meet the

assumptions of our geometrical arguments. We could then quantify (i) the similarity and broadness of empirically measured functions, and (ii) the response diversity and biomass scaling of key global change factors (Fig. 5).

### Mismatches between functions

In theory, the proportion of qualitative response mismatches between two linear ecosystem functions directly depends on their collinearity (Eq. (1)); that is, the angle between their respective directions in phase-space (the high-dimensional space whose axis reports the biomass of all constituent species). This is confirmed by our simulations, whose outcomes are represented in Fig. 2E. This basic result, however, hinges upon the assumption that perturbations are unbiased at the population level; meaning that approximately half the species show positive responses and half the species show negative responses to any given perturbation. If population-level responses are biased towards positive

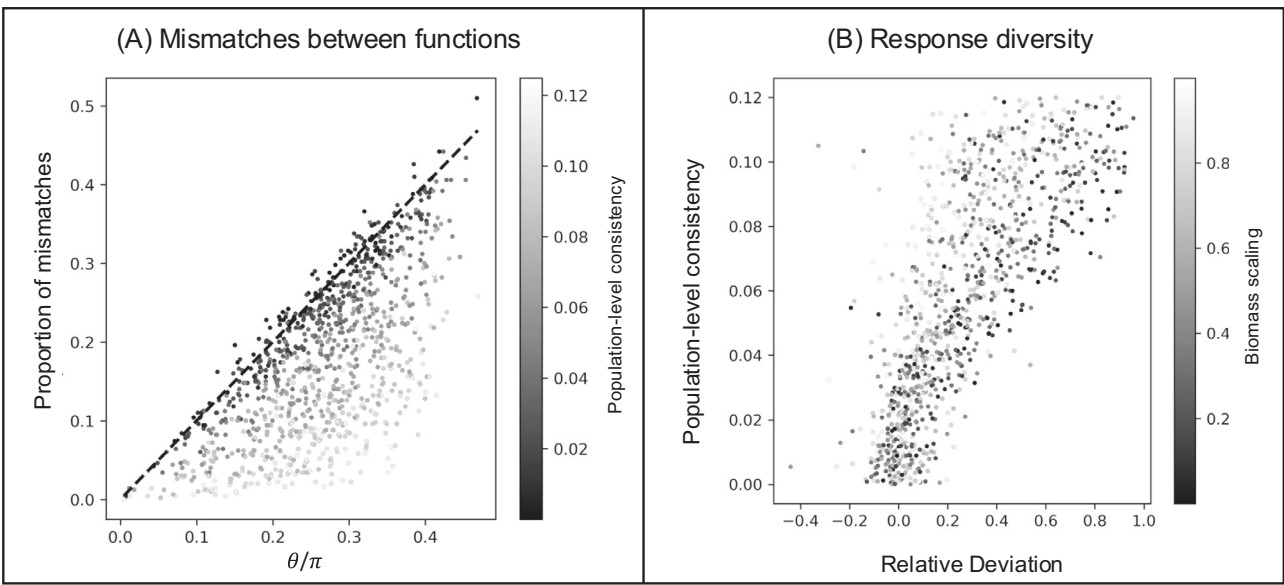

**Fig. 3 | Mismatches between functions in their responses to perturbations can be related to the response diversity of the underlying community. A** The proportion of mismatches between functions can be predicted by the angle between them in radians ($\theta$) divided by the number $\pi$ only when perturbations are unbiased at the population level. If, however, there is population-level consistency in responses to the perturbations then there are fewer mismatches than predicted as perturbations tend to fall in the mostly positive or mostly negative areas of state space, which happen to overlap with the zones of consistent observations for linear functions. **B** A notion of the response diversity of a perturbation, the consistency of population-level responses, can be estimated from the relative deviation from our baseline expectation Eq. (1).

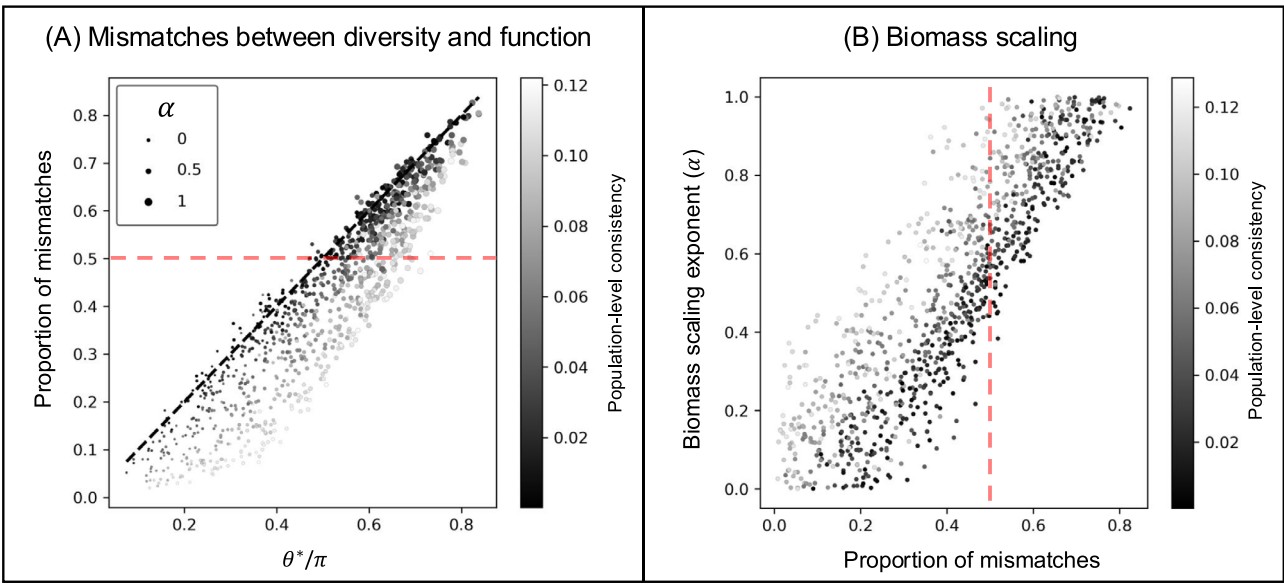

**Fig. 4 | Mismatches between functions and diversity in their responses to perturbations can be related to the biomass scaling of those perturbations. A** The angle between total biomass and the gradient of a diversity index (here $q = 2$) predicts the proportion of mismatches between them. Specifically, the relevant angle is between total biomass and the gradient of diversity after they have been scaled by the biomass of each species ($\theta^*$). When perturbations are scaled by species biomass (scaling exponent $\alpha > 0$) total biomass and diversity can effectively become opposite functions. Points above the dashed red line showcase where there is a systematic mismatch in the observations of total biomass and diversity. **B** How much a perturbation is scaled by biomass can be estimated from the proportion of mismatches between function and diversity.

or negative, the geometrical prediction overestimates the proportion of mismatches (Fig. 3). This effect occurs because when perturbation effects on species are mostly negative (or mostly positive), they tend to fall in the areas of phase space where functions will necessarily observe the same responses (top right and bottom left quadrants in Fig. 2D). Because this systematic overestimation indicates that a key assumption is violated, it informs us about population-level effects of perturbations. We can therefore deduce a link between mismatches in observations at the community level and information on population-level response diversity (Fig. 3B). Deviations from our predictions reveal a degree of population-level response diversity to the perturbation considered.

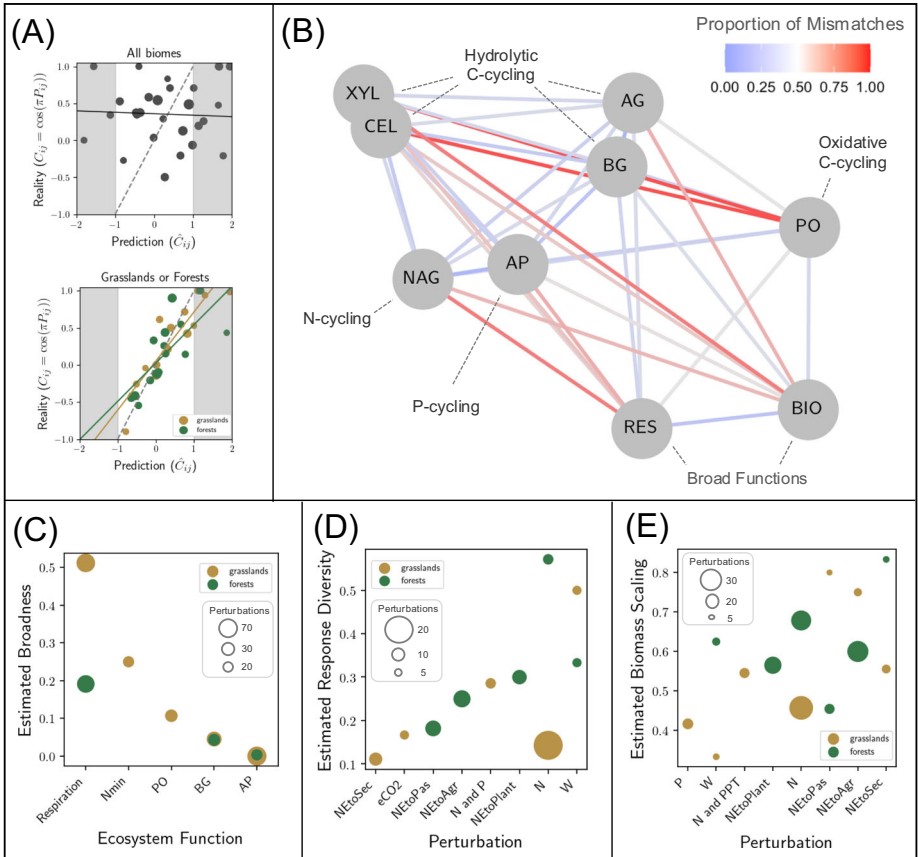

**Fig. 5 | Reanalysis of the empirical data. A** Results of the validation test, formally described in Supplementary Note 2, for the entire dataset (top) and for the grasslands data in orange and forest data in green (bottom). **B** Force-directed network constructed with the Kamada–Kawai path-length cost-function using the matrix of mismatches from the grasslands dataset as the adjacency matrix. The colour of the edges corresponds to the proportion of mismatches. The functions are microbial respiration ("RES"), microbial biomass ("BIO"), phenol oxidase ("PO"), $\alpha$-1,4-glucosidase ("AG"), $\beta$-1,4-glucosidase ("BG"), cellobiohydrolase ("CEL"), $\beta$-1,4-xylosidase ("XYL"), N-acetyl-$\beta$-glucosaminidase ("NAG"), and phosphatase ("AP"). **C** Estimates for the broadness based on mismatches between total biomass for five functions: microbial respiration, net N mineralization rate, phenol oxidase ("PO"), $\beta$-1,4-xylosidase ("XYL"), and phosphatase ("AP"). The size of the points corresponds to the number of observations that the proportion of mismatches is based on. **D** Estimates for the response diversity of perturbations based on the mismatches between total biomass and respiration (two functions with many shared observations). **E** Estimates for the biomass scaling of perturbations based on the mismatches between total biomass and Shannon diversity (the function and diversity metric with most shared observations). For parts (**D**, **E**) the size of the points corresponds to the number of observations and the perturbations are: elevated carbon dioxide ("eCO2"), nitrogen addition ("N"), phosphorus addition ("P"), warming ("W"), elevated precipitation ("PPT"), and the conversion of native ecosystems to secondary ecosystems ("NEtoSec"), to pasture ("NEtoPas"), to plantations ("NEtoPlant"), or to agriculture ("NEtoAgr").

## Mismatches between functions and diversity metrics

The probability of mismatches between ecosystem functions and diversity metrics can be predicted by considering the angle between the function and the gradient of the diversity metric (Fig. 4A). Again, consistency of responses at the population level causes the prediction to overestimate the actual proportion of mismatches. We note, however, that the angle between the direction associated with a positive function and a diversity metric can exceed 90° leading to a systematic bias towards qualitative response mismatches. This intriguing result is connected to a second piece of population-level information: the scaling of perturbations by species biomass (Box 2 Eq 7). When the effect of perturbations is larger for more abundant species, function and diversity show qualitatively different responses (only the larger points are above the red line in Fig. 4A). If a perturbation causes the biomass of abundant species to decrease, total biomass will decrease but a diversity measure related to evenness will increase. If on the other hand, a perturbation causes the biomass of abundant species to increase, total biomass will increase but evenness will decrease. This means that the degree of scaling of species responses to perturbations by their biomass can be predicted based on the observed proportion of mismatches between total biomass and diversity measures (Fig. 4B).

## Empirical results

The validation test of the geometrical framework (outlined in Supplementary Note 2) with the entire Zhou et al.[20] datasets (1235 perturbations tested across a huge diversity of biomes including agricultural systems, tundra, desert, and wetlands) was negative. We found no correlation between actual mismatches between two given functions and predicted mismatches based on the mismatches with other functions, if the latter could be seen as vectors *in a given phase space*. However, validation tests with data from either grassland systems ($n = 367$) or forest systems ($n = 435$) provided very conclusive support (strong correlation between predicted and realized mismatches) for the use of our geometrical framework (Fig. 5A). The fact that the test was inconclusive when pooling all data together should not be surprising, since the notion of unique phase space to position the different systems does not make sense. Only when grouping by biome can this fundamental assumption stand a chance of being a useful approximation (but it could very well have failed as the systems remain very different: unlike simulation experiments, the data does not represent repeated perturbations of the same system).

A network depicting the similarity of functions using the grasslands dataset further reinforced that mismatch data coincides with our

mechanistic understanding of these ecosystem functions (Fig. 5B). The only two broad functions in the network, biomass and respiration, are beside each other in the network and the seven narrow functions (production of different enzymes) grouped together as would have been expected a priori[20]. Of the ecosystem functions with enough observations to make estimates of their broadness, respiration was the broadest, followed by net nitrogen mineralization rate, and then by three specific enzymes related either to carbon cycling (phenol oxidase and beta-1,4-glucosidase) or to phosphorus cycling (phosphatase). These quantitative estimates support our basic biological intuitions about these systems: a few species contribute to the production of a specific enzyme, more species are involved in the mineralization of nitrogen, and more species still contribute to whole ecosystem respiration. The estimates of broadness for beta-1,4-glucosidase and phosphatase were almost identical for forests and grasslands but respiration was estimated to be more broad in grasslands than in forests (Fig. 5C). In Supplementary Note 3 we further show that the estimated broadness of functions based on their mismatches with total biomass can even be used to predict their actual proportion of mismatches.

The perturbations in the dataset that had enough observations for us to examine their response diversity and/or biomass scaling were warming, carbon dioxide enrichment, phosphorous addition, nitrogen addition, phosphorus and nitrogen addition combined, nitrogen addition and increased precipitation combined, and four types of land-use change: conversion from native ecosystems to agriculture, to pasture, to plantation or to secondary ecosystems. For grasslands, conversion to secondary ecosystems or the addition of nitrogen or carbon dioxide had relatively low response diversity while warming had relatively high response diversity. For forests, land-use change (particularly conversion to pastures) had relatively low response diversity while warming and the addition of nitrogen had relatively high response diversity (Fig. 5D). For grasslands, land-use change typically showed strong biomass scaling while nutrient enrichment and warming showed relatively weak biomass scaling. For forests, conversion to pasture had relatively low biomass scaling while conversion to secondary ecosystems had relatively high biomass scaling with the other perturbation types falling in between the two (Fig. 5E). In general, land-use change perturbations had low response diversity and high biomass scaling indicating that species respond in the same way (presumably negatively) and have absolute changes relative to their biomass (e.g. consistent with a perturbation decreasing 50% of all species). Perturbations like warming and nutrient enrichment, on the other hand, typically had high response diversity and low biomass scaling indicating that some species responded negatively while some responded positively to these perturbations and that absolute responses were not completely proportional to initial biomass.

## Discussion

Variability of results, or "context-dependency", is pervasive in ecology[29]. While this is partly what makes ecosystems so fascinating to study—indeed there is great interest in the mechanistic underpinning of contrasting responses of diversity and function to perturbations[30]—it could also be viewed as an obstacle to the synthesis of previous results and to the prediction of future impacts. Our research has focused on some of this variability (the variability between the responses of community aggregate properties to a given perturbation) and found that it is predictable and also a rich source of information. Mismatches between the responses of different aggregate properties to a class of perturbations (e.g. land-use change) can give us previously hidden information about the aggregate properties themselves (i.e. similarity and broadness of ecosystem functions) and about how such perturbations impact the species that constitute the community (i.e. response diversity and biomass scaling).

Ecological research is typically reductionist, using information about individuals and populations to understand communities and ecosystems[31]. Our work demonstrates the reverse approach by using information about communities to understand population-level responses.

In this paper we have reported two analyses of the microbial soil system dataset: (i) an initial, naive synthesis that we used to motivate our work (Fig. 1), and (ii) a more detailed analysis informed by our geometrical framework (Fig. 5). Our geometrical approach helped to explain some of the interesting patterns in the initial analysis—such as the relatively high levels of mismatches between broad and narrow functions and between functions and diversity—but more importantly, it allowed us to take our biological interpretations further and to extract new information from the data using a novel type of analysis. For instance, we found that perturbations associated with global change vary greatly in their response diversity (Fig. 5D). Land-use change typically had relatively low response diversity (i.e. most species responded in the same direction), while warming showed relatively high response diversity (i.e. some species increased in abundance while others decreased in abundance). Furthermore, we found that biomass scaling is a prominent feature of anthropogenic perturbations of these ecosystems. The proportion of mismatches between total biomass and Shannon diversity—positively correlated with the biomass scaling exponent (Fig. 4B)—ranged from ~0.3 for warming in grassland systems all the way up to ~0.8 for some land-use change perturbations. In other words, species that initially represent a large proportion of the overall biomass in these microbial systems also represent a large proportion of the variation in biomass caused by global change factors. In the next two sections, we will first outline in more detail the general empirical applications of our proposed framework and we will then discuss the future research directions that our geometrical perspective of aggregate properties could lead to.

## Empirical applications

Although we have shown that variability of community-level responses to perturbations can be predicted, our geometrical framework does not attempt to predict how specific aggregate properties will respond to specific perturbations. Instead, based on the assumption that functions can be seen as directions (which amounts to assuming that per-capita contributions of species to functions are fixed), it can be used to generate null expectations for when aggregate properties should and shouldn't respond in the same way to a perturbation. From a practical perspective, our framework therefore offers a novel set of methods (demonstrated in the tutorial available at: https://jamesaorr.github.io/community-properties-tutorial/) that ecologists can use to study species' contributions to ecosystem functions and the population-level effects of perturbations. As the central ingredient of our framework is the *proportion* of mismatches in the observations of different aggregate properties, increasing the volume of data will lead to more robust estimates. Indeed, the size of the points in Fig. 5C–E indicates the confidence of those estimates. However, there is a trade-off between the volume of data used to quantify proportions of mismatches and the consistency of the underlying systems; the validation test was inconclusive when we pooled data from all biomes but gave very convincing results when we focused on either the grasslands or forests systems. Given the requirements for moderate to high volumes of data, our framework is probably best suited for use in research synthesis, where it can be used to complement traditional tools like meta-analyses.

The geometrical view of aggregate properties allows us to use perturbations as probes to better understand how species influence the functioning of ecosystems. We found that the proportion of mismatches between functions can be used to quantify their similarity in terms of which species contribute to them. This was

demonstrated by the modularity of the heat map in Fig. 1A and of the network in Fig. 5B. This observation is certainly reassuring, as it confirms that mechanistic understandings at the chemical level of microbial functions are consistent with ecosystem-level observations. Furthermore, given that total biomass is by definition the broadest function, we can now use mismatches between a function of interest and total biomass to quantify the broadness of that function (demonstrated by numerical evidence in Fig. 2E inset and empirical evidence in Fig. 5C). If an ecologist was interested in a new ecosystem function they could quickly compare it to other functions based on how it responds to perturbations to estimate how broad it was and to identify which species were contributing to it (based on it's similarity to functions with more information about their species' contributions). Understanding the links between community composition and functioning has far-reaching implications for many sectors including ecosystem management, agriculture, forestry and medicine[32–34] and our approach contributes to recent efforts to study ecosystem functions in their natural context, in contrast to the traditional reductionist approach of using controlled experiments where populations or even organisms are studied in isolation[35,36].

Our framework can also be used to study population-level responses to perturbations from the top down by comparing the observations of different functions. Response diversity—the variation between species responses to a perturbation—can be measured in different ways and is a key mechanism underlying ecological stability and the biological insurance hypothesis[37–39]. Although the information we can gain using our geometrical approach (i.e. the proportion of species responding positively or negatively—see Fig. 2) is a coarse measure of response diversity, it can be accessed by just comparing the observations of different functions (e.g. total biomass and respiration) rather than actually measuring each species' response. The easiest approach is to take two functions and compare their proportion of mismatches over different perturbations (or different systems or different contexts) to gain a relative measure of response diversity (as we did in Fig. 5D). However if the collinearity between two functions is known (for well-studied functions, or by using our approximations based on the estimated broadness of the functions), then we can use the deviations from our null expectation to quantify the population-level response diversity (Fig. 3B).

Another useful piece of information that can be gained with our top-down approach is the biomass scaling of a perturbation (i.e. whether the direct effect of a perturbation is proportional to the biomass of each species). This feature of perturbations controls the relative importance of rare or common species in determining the community's temporal variability ("environmental perturbations" *sensu* Arnoldi et al.,[40]). Using mismatches between any function and any diversity metric can be used to rank perturbations based on their biomass scaling (Fig. 5E). Furthermore, the proportion of mismatches between diversity and total biomass is actually a very good proxy for the biomass scaling exponent itself Fig. 4B). If the responses of each species to the perturbations is available then biomass scaling (and response diversity) can be extracted from the data directly. However, it is very common for measures of diversity to be estimated from data without measuring species-level responses (there are 221 observations in the Zhou et al.,[20] dataset where OTU richness is the only measure of diversity). In these cases, biomass scaling cannot be measured directly from the data, but it can be estimated using our framework. Comparing multiple community-level observations—measuring responses of more functions allows for more pairwise comparisons and therefore more detailed insights—allows us to describe these features of perturbations without ever having to collect information directly at the population level, which could therefore be an efficient and cost-effective tool for research synthesis or the analysis of biomonitoring data.

## Future directions

Our work has so far overlooked the temporal dynamics of responses to perturbations. As we only needed to consider the initial and perturbed states of ecosystems for our geometrical approach (perturbations as displacement vectors in Fig. 2A), we haven't made the distinction between press and pulse perturbations and we also haven't considered non-linear responses. However, to consider a community's trajectory during and after a perturbation, our framework could be applied in future studies to test if response diversity and biomass scaling of perturbations change over time. Practically this would involve comparing the responses of two (or more) aggregate properties to a perturbation over time and checking if there was a change in the proportion of mismatches (e.g. over-replicates in an experimental treatment). For example, if the proportion of mismatches between a set of ecosystem functions was initially very low following a perturbation but then increased over time, this would be consistent with a scenario where most species initially responded negatively to that perturbation but then some species increased in abundance (e.g. due to competitive release). Changes in the proportion of mismatches between diversity and function over time would likewise imply changes in the biomass scaling of a perturbation. It seems likely that this new geometrical perspective could be combined with tools in the ecological stability literature[30] to study dynamic ecological responses to perturbations.

In our work, we did not explicitly consider biotic interactions, yet they nonetheless play a role. The state that an ecosystem reaches after a perturbation undoubtedly depends on species interactions, especially if the time scale considered is long enough to allow community dynamics to play out. The classic example is the trophic cascade[41]. If a perturbation directly impacts the top of a food chain (e.g. species invasion), it will in time also affect its base, following the alternating sign pattern characteristic of a cascade. Put in the context of our work, biotic interactions play a role in what we call "features of perturbations" like biomass scaling and response diversity. An exciting future direction would therefore be to seek for recognizable signatures of species interactions[42] in the variations through time of those perturbation features. For instance, we can hypothesise that strong mutualistic interactions would generate increasingly coherent responses as time grows (corresponding to a reduction of response diversity). Furthermore, our framework makes the simplifying assumption that species per capita contributions to functions are fixed, but in reality how a species contributes to a function may be dependent on its interactions with other species (although the fact that our validation test was conclusive implies that this assumption is not a bad approximation). We propose to see our work as a first step of a more general program: using perturbations as "probes", where ecosystem functions are macroscopic "observables", to better understand the dynamics of natural ecosystems.

Given the generality of our framework, our work touches many areas of contemporary ecology. For multifunctional ecologists, it helps to explain how different functions can respond in different ways to global change[28]. For ecologists interested in multiple perturbations, our work can be used to understand variability in how community-level properties observe the interactions (antagonistic or synergistic) between perturbations[43]. For biodiversity-ecosystem functioning research, the opposing responses of diversity and function to perturbations (which we explained) should be considered when understanding how perturbations influence biodiversity-ecosystem functioning relationships[44]. Our work can be used in disturbance ecology to link studies across disparate systems[45] and may even help to interpret trade-offs between biodiversity and crop yield under different farming practices[46]. When studying complex systems such as ecosystems, it is important to have baseline expectations for their behaviour. We have found that the variability between community-level responses to perturbations does not just limit synthesis and

## BOX 2

# Formalizing the variability of observed responses to a perturbation

We formalize the process of observing the ecosystem-level impact of a given perturbation, based on aggregate features of functioning or diversity. Our goal is to explain what controls the probability that two scalar observations of the same perturbed ecosystem give opposite results. Here bold symbols denote $S$—dimensional vectors, where $S$ is the species richness of the community. Let $\mathbf{N}^c$ be the initial (or control) state of a community: the vector of species biomass prior to the perturbation. Let $\mathbf{N}^p$ be the perturbed community state. The observed response, quantified via an ecosystem function $f(\mathbf{N})$, is

$$\Delta f = f(\mathbf{N}^p) - f(\mathbf{N}^c). \qquad \text{(Box 2 Eq 1)}$$

For a linear function, there exists a constant $f_0$ (because we will consider changes in functioning, and not absolute levels of functioning, this constant will play no role in what follows) and a vector $\boldsymbol{\varphi}$—the gradient—such that

$$f(\mathbf{N}) = f_0 + \langle \boldsymbol{\varphi}, \mathbf{N} \rangle \qquad \text{(Box 2 Eq 2)}$$

with $\langle \cdot, \cdot \rangle$ the scalar product of vectors. The elements of the gradient vector $\boldsymbol{\varphi}$ encode the per capita contribution of species to the function. For us it will not matter what those exact contributions are. Only relative species contributions, which determine the *direction* spanned by the vector $\boldsymbol{\varphi}$, are required for our framework. A positive function is such that the elements of the gradient are positive. If we rewrite the response of the function to the perturbation, we get that

$$\Delta f = \langle \boldsymbol{\varphi}, \Delta \mathbf{N} \rangle \qquad \text{(Box 2 Eq 3)}$$

where $\Delta \mathbf{N} = \mathbf{N}^p - \mathbf{N}^c$ is the vector of population-level responses. For non-linear aggregate properties, such as diversity metrics, the (state dependent) gradient vector can be computed as $\varphi_i(\mathbf{N}^c) = \frac{\partial f}{\partial N_i} \big|_{\mathbf{N}^c}$. In this case, expression (Box 2 Eq 3) will be an approximation, accurate for weak perturbations for which the state-dependent gradient vector is still relevant. Now, for two functions, $f, g$ associated with two directions spanned by the two gradient vectors $\boldsymbol{\varphi}$ and $\boldsymbol{\phi}$, we define their collinearity as the angle $0 \leq \theta < 2\pi$ whose cosine is

$$\cos \theta = \frac{\langle \boldsymbol{\varphi}, \boldsymbol{\phi} \rangle}{\|\boldsymbol{\varphi}\|\|\boldsymbol{\phi}\|} \qquad \text{(Box 2 Eq 4)}$$

where $\|\cdot\|$ denotes the Euclidian norm of vectors. A graphical argument (Fig. 2D) tells us that the fraction of perturbation vectors $\Delta \mathbf{N}$ that will lead to a mismatch between the observations of $f$ and $g$ is

$$\mathbb{P}(\text{sign}(\Delta f) \neq \text{sign}(\Delta g)) = \frac{\theta}{\pi} \qquad \text{(Box 2 Eq 5)}$$

In such cases, one of the functions will observe a positive response, while the other function will observe a negative response. Generically, we can evaluate the cosine of the angle based on a notion of functional broadness. Indeed, given a random choice of positive functions

$$\frac{\langle \boldsymbol{\varphi}, \boldsymbol{\phi} \rangle}{\|\boldsymbol{\varphi}\|\|\boldsymbol{\phi}\|} \approx \frac{1}{S} \frac{\sum \varphi_i \sum \phi_i}{\sqrt{\sum \varphi_i^2 \sum \phi_i^2}} = \frac{1}{S} \sqrt{\frac{1}{\sum \left(\frac{\varphi_i}{\sum \varphi_i}\right)^2} \frac{1}{\sum \left(\frac{\phi_i}{\sum \phi_i}\right)^2}} = \sqrt{\frac{{}^2 D_f}{S} \frac{{}^2 D_g}{S}} \qquad \text{(Box 2 Eq 6)}$$

where ${}^q D$ denotes Hill's diversity index. We will call the fraction $\frac{{}^2 D_f}{S}$ the *broadness* of the function $f$, which is maximal (and equal to one) if all species contribute equally to the function (i.e. total biomass).

We can modify the above theory to account for an additional piece of population-level information in the form of a biomass scaling of population-level responses. It is indeed reasonable to expect that more abundant species will, in absolute terms, show a larger response to some types of perturbations (e.g. habitat loss of 50% may decrease biomass of all species by 50%, so the most abundant species will experience the greatest absolute losses). For some scaling exponent $\alpha \geq 0$, if we denote $\Lambda$ the diagonal matrix whose elements are the species biomass prior to the perturbation, we may assume that the perturbation displacement vector takes the form $\Delta \mathbf{N} = \Lambda^\alpha \boldsymbol{\Delta}$. We then have that

$$\Delta f = \langle \Lambda^\alpha \boldsymbol{\varphi}, \boldsymbol{\Delta} \rangle \qquad \text{(Box 2 Eq 7)}$$

the relevant angle to consider then becomes

$$\cos \theta_\alpha = \frac{\langle \boldsymbol{\varphi}, \Lambda^{2\alpha} \boldsymbol{\phi} \rangle}{\|\Lambda^\alpha \boldsymbol{\varphi}\|\|\Lambda^\alpha \boldsymbol{\phi}\|} \qquad \text{(Box 2 Eq 8)}$$

giving the fraction of rescaled vectors $\boldsymbol{\Delta}$ that would lead to a qualitative mismatch.

prediction in ecology. Instead, this variability is predictable and can be leveraged to gain useful information about species' responses to perturbations and species' contributions to ecosystem functioning. Our work provides a solid platform from which the complexity of community-level responses to anthropogenic global change can be better understood.

## Methods

### Geometrical approach

To understand what can be learned from the variability of aggregate properties' responses to perturbations, we transpose the ecological problem to a more abstract, but simpler, geometrical setting (described more formally in Box 2).

First, we consider the effects of perturbations on populations as displacement vectors in the ecosystem's state-space, where axes report the biomass of all constituent species (Fig. 2A). This vector is the difference between initial and perturbed states. It encodes the response to the perturbation at the population level at a given time and can be applied to both press perturbations (where the community may be expected to stay at the perturbed state for some time) and pulse perturbations (where the community may be expected to return to the initial state from the perturbed state). We then see ecosystem functions as positive directions in this same state space (Fig. 2B). Total biomass for example is the sum of all the species' biomass and its direction lies exactly between all the axes, giving equal weight to all species. Other functions may not be influenced by the biomass of all species equally. In the hypothetical example shown in Fig. 2B, general decomposition is slightly more sensitive to the biomass of fungi than to the biomass of bacteria, plastic decomposition is primarily carried out by bacteria, and chemical production is primarily carried out by fungi. In general, a positive direction is spanned by a vector of positive values representing the per-capita contribution of each species to the function of interest. Our approach therefore aligns with Grime's "biomass-ratio hypothesis" where species contributions to ecosystem functions increase with increasing biomass[47]. The "broadest" function, total biomass, is made up entirely of ones. The "narrowest" functions, are made up entirely of zeroes, except on the entry associated with the only contributing species[24].

Next, we combine these two levels of abstraction to model how functions "observe" perturbations. We recenter the state space so that the axes now represent the response of each species, with the origin consequently being the initial state of the community (Fig. 2C). Projecting the displacement vector (multi-dimensional vector describing species responses to a perturbation) onto the direction of an ecosystem function (one dimensional vector made up of species contributions to the function) gives the "observation" of that function (see blue and red lines coming from perturbed states A and B in Fig. 2C). For each function, drawing a line through the origin and perpendicular to the direction of the function delineates two zones. One where the projection is negative, and thus the function observes a negative response and the other where the projection is positive and thus the function observes a positive response. If the two directions associated to the two functions are not perfectly collinear, there will be zones of state-space where responses to perturbations will be qualitatively different when observed by one function or the other. These zones are the two symmetrical cones centred on the origin, formed by the delineation lines of the functions, perpendicular to their respective directions (red zones in Fig. 2D). The larger the angle between two functions, the larger the zones of mismatches. Consequently, if species' responses were random and unbiased, the probability of finding a qualitative mismatch between two functions is:

$$\mathbb{P}(\text{Mismatch}) = \frac{\theta}{\pi} \tag{1}$$

where $\theta$ is the angle between the two functions measured in radians. This collinearity of functions allows us to quantify their similarity. The similarity between functions, defined in this way, is related to their respective broadness, which quantifies the evenness of species per-capita functional contributions (Box 2 Eq 6). Indeed, in a community of $S$ species and functions $f$ and $g$:

$$\cos \theta \approx \cos \theta_{div} = \sqrt{{}^2D(f)/S \times {}^2D(g)/S} \tag{2}$$

where $1/S \leq {}^2D(f)/S \leq 1$ is the broadness of the function $f$ (same for function $g$), defined here as the Gini–Simpson diversity index[12] of the vector of species contributions to the function, and normalized by species richness $S$. Expression (2) quantifies the intuitive expectation that two broad functions ought to be highly collinear, whereas two narrow functions can be independent (i.e. orthogonal to one another) if they are not performed by the same set of species.

There is a straightforward, yet very useful application to this reasoning that we will use in our data analysis: because total biomass is the broadest function by definition (corresponding to a value of 1), we can use the proportion of mismatches $P_{f,bio}$ between total biomass and a given function $f$ to estimate the latter's broadness. Indeed, if perturbations are random, we have, for any positive function:

$$\cos(P_{f,bio} \times \pi)^2 = {}^2D(f)/S \tag{3}$$

We illustrate this relationship between broadness and mismatches with total biomass in Fig. 2E (inset).

Our final level of abstraction is the realization that measures of diversity, which are highly non-linear functions of species biomass (in the mathematical sense of a function of variables, not in the sense of ecological functioning), can still be placed into this geometrical setting by considering their (state-dependent) gradients (outlined in more detail in Box 2). The gradient of a diversity metric is a state-dependent vector encoding how small variations in each species' biomass change that diversity metric. The collinearity between diversity metrics and ecosystem functions can therefore be quantified by measuring the angle between the gradient of a diversity metric and the direction of an ecosystem function. Importantly, gradients of diversity metrics span non-positive directions in state space because increasing the biomass of some species (the more abundant ones) decreases diversity. This allows for the angle between diversity metrics and ecosystem functions to exceed 90°.

### Simulation model for perturbation experiments

To test, explore and illustrate the geometrical ideas outlined above, we conducted numerical experiments where ecological communities were perturbed and their responses were observed using different aggregate properties. We did not ask our simulations to have complex, realistic underpinnings. We simply defined a protocol to generate a wide range of initial and perturbed states, and a wide range of aggregate properties (representing ecosystem functions or diversity measures) that we then used to quantify the ecosystem-level impacts of the perturbations.

Initial states were vectors $N$ of length $S$ (chosen uniformly between $S = 20$ and $S = 100$) whose elements $N_i$ are the initial species abundance or biomass. Those were drawn from log-normal distributions with zero mean and standard deviation (uniformly chosen between 1/2 and 2), thus generating a wide range of communities while also mimicking empirical abundance distribution patterns. For each initial state, 500 perturbations were generated as vectors $\Delta N$ of length $S$ (perturbed states are $N + \Delta N$) whose elements $\Delta N_i$ were generated in the following way. First, for each species, we drew a value $x_i$ from a normal distribution with unit standard deviation and mean $\mu$. For a given initial state, $\mu$ is a fixed value uniformly chosen between −0.3 and

0.3. It determines the qualitative consistency of population-level responses (more on this below). We then normalized the set of values $x_i$ by $(\frac{1}{S}\sum x_j^2)^{1/2}$, which gave us a set of values $y_i$ that we used to define the actual response of species as

$$\Delta N_i = \text{intensity} \times y_i \times N_i^\alpha \qquad (4)$$

For a given perturbation, its intensity was drawn uniformly between 0 and 0.1. We also allowed the impacts of perturbations to scale with the initial abundance (or biomass, in this toy model there is no difference) of species. For each perturbation, the biomass scaling exponent ($\alpha$) was uniformly chosen between 0 and 1. When $\alpha = 1$, the population response to the perturbation is, on average over the community, proportional to the species initial biomass. The other basic population-level feature that we considered is a notion of response diversity (i.e. whether the perturbation impacted most species positively or negatively). As mentioned above, this feature is set by the parameter $\mu$. Indeed, if we define the population-level response consistency as

$$\text{bias} = |\frac{1}{2} - \frac{\#\{i, \Delta N_i < 0\}}{S}| \qquad (5)$$

(# denotes the number of elements in a discrete set, here the set of species whose abundances are reduced by the perturbation); then, the expected fraction of negative population responses in the above expression is $\Phi(-\mu)$ where $\Phi(x)$ is the cumulative function of a standard normal distribution.

Ecosystem functions, which we used to "observe" the ecosystem-level response to perturbations, were represented by positive directions in an $S$-dimensional space, spanned by vectors $\boldsymbol{\varphi}$ whose elements $\varphi_i$ represent species' per-capita functional contributions. For a given state $\boldsymbol{N}$, its level of functioning is then $f(\boldsymbol{N}) = \sum \varphi_i N_i$ (see Box 2). The per-capita contributions $\varphi_i$ were drawn from a log-normal distribution with a standard deviation uniformly chosen between 0 and 1.3. When the standard deviation was small, the functions were broad as the per-capita contributions of each species were similar. When the standard deviation was large, however, the functions were more narrow, with a large variation in the per-capita contributions of each species to the function.

Diversity metrics were taken from the family of Hill diversity that define the effective number of species as:

$$^qD(\boldsymbol{N}) = \left(\sum_{i=1}^{S} p_i^q\right)^{1/(1-q)} \qquad (6)$$

where $S$ is richness, $p_i$ is the relative abundance (or biomass) of species $i$ and $q$ is the hill number that determines the sensitivity of the diversity index to rare or to abundant species. This general equation encompasses species richness ($q = 0$), the Shannon index ($q = 1$) and the Gini−Simpson index ($q = 2$)[9,10,12,22]. To apply our geometrical framework to diversity observations we considered the directions spanned by their gradients (the vector of partial derivatives $\frac{\partial ^qD}{\partial N_i}$), evaluated at the initial state, which take the form $^q\varphi = (^q\varphi_i)$ with

$$^q\varphi_i = \frac{q}{1-q}\left(p_i^{q-1} - \sum_{j=1}^{S} p_j^q\right) \qquad (7)$$

For each perturbation experiment and each pair of aggregate properties $f$, $g$−either two positive linear functions, or a diversity metric and a function (for two diversity metrics see Supplementary Note 1)−we checked the consistency of their responses. That is, we looked at the sign of $f(\boldsymbol{N}+\boldsymbol{\Delta N}) - f(\boldsymbol{N})$ and compared it to the sign of $g(\boldsymbol{N}+\boldsymbol{\Delta N}) - g(\boldsymbol{N})$. If they do not coincide, there is a qualitative mismatch between the two ways of observing the ecosystem's response to

the perturbation. For the simulations, 1000 communities (i.e. initial states) were generated and each one experienced 500 different perturbations. For Figs. 2E and 3, two ecosystem functions of varying broadness were generated for each community and used to observe the community-level responses to the perturbations. The angle between the directions defined by the functions was calculated, divided by $\pi$ (Eq. (1)), and plotted against the realised proportion of mismatches over the 500 perturbations, while recording the relative deviation from the prediction. For Fig. 2E all perturbations were unbiased at the population level, but for Fig. 3 perturbations could vary in their population-level consistency. The angle between each pair of functions was also estimated using only the knowledge of their broadness based on their mismatches with biomass. For Fig. 4A, total biomass (positive direction whose elements are all 1) and Hill-Simpson ($^2D$) were used to observe the ecosystem-level responses to the perturbations. The effective angle between total biomass and the state-dependent gradient of the diversity index, based on (Box 2 Eq 8), was calculated, divided by $\pi$, and plotted against the actual proportion of mismatches.

### Detailed analysis of empirical data

Equipped with our geometrical framework for understanding the variability of functional and biodiversity responses to perturbations we can return to the empirical data from Box 1 to uncover novel insights. However, before we can use our framework to learn more about species contributions to ecosystem functions and about the structure of perturbations, we can first confirm that viewing functions as directions and equating their mismatches to their collinearity is a valid approach for a given dataset. To do this we can perform a validation test, formally described in Supplementary Note 2, where we try to predict the mismatches between two functions (i.e. their collinearity) based on the mismatches between all other pairs of functions. Indeed, if we know the respective angles that two chosen directions make with the remaining set of directions, we should be able to estimate, *in a specific way*, the angle between the chosen pair. This test involves matrix operations that can introduce artefacts into the results, meaning that an inconclusive test does not necessarily invalidate the application of our framework to a given dataset. However, a conclusive test−mismatches between two functions being well predicted by mismatches between all other pairs of functions−is very strong support for the view of aggregate properties as directions in state space and gives a green light for further exploration of the data using our geometrical arguments.

To better understand species' contributions to ecosystem functions we can use the mismatch data (i.e. matrix in Fig. 1) to examine both the similarity of functions and their relative broadness. Firstly, the matrix of mismatches can be used as an adjacency matrix for a network that groups functions based on their similarity. A force-directed layout algorithm, such as the Kamada−Kawai path length cost-function[48], will generate networks where distance corresponds to the similarity of functions. Secondly, we can use total biomass (the broadest ecosystem function by definition) as a baseline to quantify the broadness of other functions. The angle between total biomass and other broad functions will be small so, over many perturbations with unbiased population-level effects, the proportion of mismatches will therefore be low. Narrower functions will have larger angles with biomass, which will result in higher proportions of mismatches (Fig. 2). Working in reverse, we can use the proportion of mismatches between some function and total biomass (directly available from the data) to predict the broadness of that function. For a fair estimate of broadness, the proportion of mismatches between the function and biomass should be quantified over a large pool of perturbations that collectively have random effects. Here, we therefore do not consider perturbations of nutrients for ecosystem functions related to that nutrient−these perturbations have systematic effects rather than random effects−and we only

consider cases where the proportion of mismatches between a function and biomass is based on at least twenty perturbations including at least five types of global change factors.

So far we have used perturbations to gain insights into species contributions to ecosystem functions. However, we can also use the mismatches between functions to gain useful information about the population-level effects of the perturbations themselves. We can compare the proportion of mismatches between two ecosystem functions (e.g. total biomass and respiration) across different perturbations to quantify the relative response diversity of those perturbations. If perturbations have low response diversity (i.e. most species respond in the same direction), then perturbations will be biased in their directions in state-space towards the fully negative or fully positive areas of state-space (bottom left quadrant or top right quadrant of Fig. 2D, respectively), and would avoid the cones of mismatches for functions with positive directions. We can therefore use the proportion of mismatches for a given pair of functions to rank perturbations based on their response diversity. We can also use mismatch data to ask if a perturbation's population-level effects are independent of biomass or if more abundant species have larger absolute changes in biomass (i.e. biomass scaling of a perturbation). If a perturbation causes the biomass of abundant species to decrease, total biomass will decrease but diversity will increase. If on the other hand, a perturbation causes the biomass of abundant species to increase, total biomass will increase but diversity will decrease. As such, when perturbations are scaled by biomass, there will be a higher proportion of mismatches between functions and diversity. As a result, we can use the proportion of mismatches between a function and a diversity metric to rank perturbations based on their biomass scaling. Here, we only made estimates for the response diversity or biomass scaling of perturbations if there were at least five shared observations of those perturbations for the relevant pair of aggregate properties in the dataset.

A detailed tutorial, aimed at empirical ecologists interested in applying this geometrical framework to their data, is available at https://jamesaorr.github.io/community-properties-tutorial/. The tutorial contains useful snippets of code and detailed descriptions of all stages of the analysis from (i) data preparation, (ii) validation test, (iii) exploring species contributions to functions, and (iv) exploring the population-level effects of perturbations.

## Reporting summary

Further information on research design is available in the Nature Portfolio Reporting Summary linked to this article.

## Data availability

The data used and generated in this study are available at https://doi.org/10.5281/zenodo.13985015[49].

## Code availability

The code used in this study is available at https://doi.org/10.5281/zenodo.13985015. A detailed tutorial with example code is available at https://jamesaorr.github.io/community-properties-tutorial/.

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

## Acknowledgements

J.A.O. and M.C.J. were supported by a Natural Environment Research Council grant (NE/V001396/1). J.A.O. was supported by an Irish Research Council Laureate Award (IRCLA/2017/112) and the TCD Provost's PhD Award held by J.J.P. J.-F.A. and A.L.J. were supported by an Irish Research Council Laureate Award (IRCLA/2017/186). J.-F.A. was supported by the TULIP Laboratory of Excellence (ANR-10-LABX-41).

## Author contributions

J.A.O. and J.-F.A. conceived the original idea. J.-F.A. led the development of the theoretical framework with input from J.A.O. J.A.O. analysed the data, conducted the simulations, developed the tutorial, and produced the figures with support from J.-F.A. J.A.O. wrote the first draft of the manuscript with support from J.-F.A. J.J.P., A.L.J., and M.C.J. provided additional support throughout the project. All authors contributed to the final manuscript.

## Competing interests

The authors declare no competing interests.
