## [Transparent Peer Review file · Nature Communications]

Variability of functional and biodiversity responses to perturbations is predictable and informative

Corresponding Author: Dr James Orr

Version 0:

Reviewer comments:

Reviewer #1

(Remarks to the Author)

Thanks for a thorough revision of the manuscript that I have been seeing at a previous stage. The authors have done a terrific job dealing with my comments. The new manuscript has much matured compared to its previous version. I laud the new focus on the empirical applicability of the concept and the tutorial provided.

(Remarks on code availability)

Reviewer #4

(Remarks to the Author)

The manuscript by Orr et al. presents a new framework for relating the variable responses of ecosystems to different environmental perturbations by representing different perturbations as vectors in the same ecological state space. Then, authors demonstrate how this approach could be used by empiricists—from testing whether this framework is applicable to a dataset, to quantifying the response diversity of perturbations and clustering similar functions via a network.

For this re-review, I worked jointly with a junior colleague, and we were especially impressed by the significant work authors did to demonstrate how their framework can be applied to empirical datasets to learn things. For me personally, this was the most pressing issue with the original submission, and I believe the revisions addressed this thoroughly. Re-reading the work in the context of my colleagues' comments, and the authors' responses, I believe the edits improved the work substantially. In the few cases where they didn't directly implement reviewer suggestions, they provided reasonable justifications for doing so, and addressed the root concerns in a different way.

The manuscript also benefitted from changes making it more consistent and clear, and having more concrete future directions. We only noticed a few minor typos that remain. In the description to Figure 2E, there is a mismatch between “Two broad functions (i)” and the figures, where “Two broad functions” is labeled by (ii). In line 206, there is a typo “If we known the respective angle.”

(Remarks on code availability)

Did not review the code.

Reviewer #5

(Remarks to the Author)

(Remarks on code availability)
